# Resistive Sensing of Seed Cotton Moisture Regain Based on Pressure Compensation

**DOI:** 10.3390/s23208421

**Published:** 2023-10-12

**Authors:** Liang Fang, Ruoyu Zhang, Hongwei Duan, Jinqiang Chang, Zhaoquan Zeng, Yifu Qian, Mianzhe Hong

**Affiliations:** 1College of Mechanical and Electrical Engineering, Shihezi University, Shihezi 832000, China; fangliang@stu.shzu.edu.cn (L.F.); dhwsg123@sina.com (H.D.); changjinqiang@stu.shzu.edu.cn (J.C.); zhaoquan_zeng@shzu.edu.cn (Z.Z.); qianyifu@stu.shzu.edu.cn (Y.Q.); hongmianzhe@stu.shzu.edu.cn (M.H.); 2Key Laboratory of Northwest Agricultural Equipment, Ministry of Agriculture, Shihezi 832000, China

**Keywords:** pressure, density, conductivity, moisture regain, sensor, machine learning

## Abstract

The measurement of seed cotton moisture regain (MR) during harvesting operations is an open and challenging problem. In this study, a new method for resistive sensing of seed cotton MR measurement based on pressure compensation is proposed. First, an experimental platform was designed. After that, the change of cotton bale parameters during the cotton picker packaging process was simulated through the experimental platform, and the correlations among the compression volume, compression density, contact pressure, and conductivity of seed cotton were analyzed. Then, support vector regression (SVR), random forest (RF), and a backpropagation neural network (BPNN) were employed to build seed cotton MR prediction models. Finally, the performance of the method was evaluated through the experimental platform test. The results showed that there was a weak correlation between contact pressure and compression volume, while there was a significant correlation (*p* < 0.01) between contact pressure and compression density. Moreover, the nonlinear mathematical models exhibited better fitting performance than the linear mathematical models in describing the relationships among compression density, contact pressure, and conductivity. The comparative analysis results of the three MR prediction models showed that the BPNN algorithm had the highest prediction accuracy, with a coefficient of determination (R^2^) of 0.986 and a root mean square error (RMSE) of 0.204%. The mean RMSE and mean coefficient of variation (CV) of the performance evaluation test results were 0.20% and 2.22%, respectively. Therefore, the method proposed in this study is reliable. In addition, the study will provide a technical reference for the accurate and rapid measurement of seed cotton MR during harvesting operations.

## 1. Introduction

Cotton is a cash crop in the world [1]. In recent years, due to the rapid increase in the cotton planting area, machine-harvested cotton has been widely adopted. However, machine picking causes a decline in cotton quality [2]. Moisture regain (MR) is an important indicator of cotton quality. MR not only affects cotton storage, transportation, processing, and textile quality; it also acts as a key factor in the cotton trade. During the harvesting operations of a cotton picker, a low MR can cause cotton fibers to break easily, while a high MR can result in a high impurity content. In the storage and transportation stage, a high MR can cause mold, rot, and even spontaneous combustion of the cotton bales [3,4,5]. During cotton transaction stage, the existing offline MR measurement methods have low efficiency, and the measurement results are often influenced by human factors. It is urgent to develop an online method by which to measure the MR of seed cotton accurately and rapidly during the harvesting operations. This can not only standardize the operation of cotton picker; it can also provide a data reference for cotton trade and cotton processing. However, the measurement of seed cotton MR during the harvesting process is challenging due to the constantly changing density during packaging. Therefore, finding a solution with which to minimize the impact of density changes on MR measurement is crucial for achieving an accurate online measurement of seed cotton MR.

There are direct and indirect methods by which to measure cotton MR. The direct methods have high accuracy but are time-consuming and inefficient. Indirect methods, known for their high speed and sustainability, have experienced rapid development [6]. Resistive sensing is one of the indirect methods used for measuring cotton MR. It offers advantages such as speed, accuracy, and cost-effectiveness. Many scholars have studied this method [7,8,9]. Akahira [10] and Byler [11] constructed a quantitative relationship between resistive values and moisture content of cotton. Zheng et al. [12] and Wang et al. [13] designed an online MR measurement device based on resistive sensing, achieving dynamic MR measurement of seed cotton in the cotton processing stage. Zhang et al. [14] proposed a rapid cotton quality measurement system that enables the fast measuring of various indices during the cotton trading stage, and the MR measurement module was based on the resistive sensing.

Previous studies of cotton MR measurement have mostly focused on the trading and processing stages of cotton, and few studies have explored the MR measurement during the harvesting operations. Cotton fibers are fluffy in their natural state and have variable density. Their density needs to be maintained within a certain range in order to achieve stable conductivity [15]. Therefore, it is necessary to study the interactions between compression density, contact pressure, and conductivity during the densification process of seed cotton. Many scholars have conducted relevant studies [16,17,18,19]. For example, Kong et al. [20] and Ren et al. [21] conducted a compression experiment on seed cotton and raw cotton to investigate the mechanical characteristics of cotton during the compression process, respectively. Guo et al. [22] studied the load measurement and automatic adjustment of bale density of a baler and constructed a relationship between sidewall load and bale density. Li et al. [23] explored the effect of textile structure on its conductivity and discussed the influence of fabric density and surface density on textile conductivity.

In summary, there is limited research on the measurement of seed cotton MR during the harvesting operations, and most studies have not considered the impact of changes in cotton density on measurement results. This greatly affects measurement accuracy. To ameliorate the difficulties in the rapid and accurate measurement of MR during harvesting operations, this study proposes a new method for the resistive sensing of seed cotton MR measurement based on pressure compensation. Specifically, densification experiments on seed cotton were conducted in order to explore the parameter changes of cotton bales during the baling process of the cotton picker. Then, the interactions between compression density, contact pressure, and conductivity of seed cotton were investigated, and algorithmic models for predicting MR were built. Finally, the performance of the MR measurement method was tested using an experimental platform. The research can provide a technical reference for the online measurement of seed cotton MR in harvesting operations, which is of great significance to the digital development of agricultural machinery.

## 2. Materials and Methods

### 2.1. Materials

Machine-harvested cotton (cultivar Xinluzao 80), harvested in Jimusaer County, Changji, Xinjiang Uyghur Autonomous Region, China (44°30′ N, 88°89′ E) in November 2022, were collected. The lint weight of cotton is 5.6 g, with a lint percentage of 43.0%. The average length of the fibers in the upper half is 30.0 mm, with a strength (tenacity) of 31.4 cN/tex and a micronaire value of 4.7. The uniformity index is 85.3% [24]. According to the moisture absorption principle of cotton fibers, cotton samples with different MR levels were prepared, and the MR calculation formula is as follows:(1)$$MR=\frac{m-m_{0}}{m_{0}}\times 100\%,$$
where *MR* is the moisture regain of the sample, %; *m* is the weight of sample before drying, g; and *m*_0_ is the weight of sample after drying, g.

First, cotton samples were dried to a low initial MR level (<6%). Second, the cotton samples were placed in the constant temperature and humidity test chamber (FQY/WSK-400C, Guangzhou Fengqianyuan Environmental Test Equipment Co., Ltd., Guangzhou, China), with a temperature deviation of ±2.0 °C and a humidity deviation of ±3.0% RH. In addition, the samples were balanced for 24 h at different temperatures and relative humidity levels to prepare cotton samples with different MR levels (Table 1). After the cotton samples reached equilibrium, 150 g and 50 g of cotton samples were separately weighed by an electronic balance (XY500C, Changzhou Xingyun Electronic Equipment Co., Ltd., Changzhou, China), with a measurement accuracy of ±0.01 g. The 150 g cotton sample was placed in the cotton compressing device for MR measurement. The 50 g cotton sample was used to measure the actual MR value according to the “Test Method for Moisture Regain of Raw Cotton—Oven Method” [25] using the semi-automatic constant-temperature oven (YG767, Nantong Sansi Electromechanical Science and Technology Co., Ltd., Nantong, China) with a temperature control accuracy of ±1 °C. This actual value was used to calibrate the MR values obtained through resistive sensing.

### 2.2. Data Acquisition

The experimental platform consisted of a computer, a microcontroller (STM32F103, Guangzhou Xingyi Electronic Technology Co., Ltd., Guangzhou, China), an aluminum frame, a 24 V DC switching power supply, a 24 V DC linear actuator (working stroke: 150 mm; maximum thrust: 1000 N), a motor driver, a digital transmitter, organic glass sample boxes with different diameters, and a seed cotton MR sensor. The cotton compression unit comprised cotton compression plates with different diameters, sensing electrodes, and a pressure sensor with a range of 0~5 kg and a full-scale accuracy of 0.3% (Figure 1).

The principle of resistive sensing of seed cotton MR based on pressure compensation is as follows: the sense probe of the seed cotton MR sensor is a planar structure, and the planar surface directly contacts the cotton fiber sample during operation (Figure 2). Previous studies have shown that cotton fibers with different MR levels have different degrees of conductivity [26,27]. Therefore, in this study, the resistance of cotton fibers was measured using the drive and sense electrode. The pressure sensor was used to measure the contact pressure, which was then used to characterize the compression density. After that, the pressure value was used as a compensation value for the resistive measurement to indirectly determine the MR of the seed cotton.

During the experiment, we adjusted the sample box holder to fit different diameter cotton sample boxes. Seed cotton was placed in the cotton sample box, and the microcontroller controlled the motor driver to drive the linear actuator to apply load on the seed cotton. The resistance of the seed cotton was measured using the bridge tester (LCR-8110G, Good Will Instrument (Suzhou) Co., Ltd., Suzhou, China), with a measurement accuracy of ±0.1%, and the instrument was calibrated using a standard resistor. The pressure was obtained using the pressure sensor and digital transmitter, and calibration was conducted using weights.

### 2.3. Experiment Methods

#### 2.3.1. Seed Cotton Densification Experiment

It was found that a portion of cottonseeds exhibited slight fragmentation when the compression density reached 250 kg·m^−3^. Therefore, the seed cotton was compressed from its natural bulk density (90 kg·m^−3^) to 250 kg·m^−3^ for the densification experiment. Five sets of 150 g seed cotton samples were weighed and placed in sample boxes with a diameter of 16 cm. A load was applied using a DC linear actuator and stopped when the seed cotton density reached 250 kg·m^−3^. In addition, the load was released after being held for 45 s. The contact pressure was recorded by the host computer. After completing the measurement of the five sample sets, the pressure–time curves were obtained.

#### 2.3.2. Factors Influencing Contact Pressure

The experiment characterizes the dynamic changes in volume during the densification of cotton fiber aggregate by altering the compression diameter and height in order to explore the impact of changes in the compression volume of cotton fiber aggregates on contact pressure (Figure 3).

The formulae for calculating the compression density of cotton fiber aggregates are as follows:(2)h=H−x
(3)$$\rho =\frac{4\times m}{\pi D^{2}h},$$
where *H* is the height of the sample box, 15 cm; *x* is the downward displacement of the cotton compressing plate, cm; *ρ* is the compression density of cotton fiber aggregates, kg·m^−3^; *m* is the weight of the cotton fiber aggregate in the sample box, kg; *D* is the inner diameter of the sample box (the compression diameter of the cotton fiber aggregate), cm; and *h* is the compression height of the cotton fiber aggregates inside the sample box, cm.

Five seed cotton samples (150 g per sample) were used for the experiment at a constant compression height and density (*h* = 5 cm; *ρ* = 200 kg·m^−3^). Sample boxes with different diameters were used (*D* = 12, 14, 16, 18, 20 cm), and corresponding cotton sample weights for the experiment were weighed. Each sample was measured five times. Similarly, at a constant compression diameter and density (*D* = 16 cm, *ρ* = 200 kg·m^−3^), the compression height was changed (*h* = 4, 5, 6, 7, 8 cm), and the corresponding cotton sample weights for the experiment were weighed. Each sample was measured five times. Finally, at a constant compression height and diameter (*h* = 5 cm, *D* = 16 cm), the compression density was changed (*ρ* = 160, 200, 240 kg·m^−3^), and the corresponding cotton sample weights for the experiment were weighed. Each sample was measured five times. After processing the experimental data, a graph illustrating the relationships between contact pressure, compression height, compression diameter, and compression density was obtained.

#### 2.3.3. Influencing Factors of Conductivity

The relationship between conductivity and resistance was analyzed, and the calculation formula between the two is as follows:(4)$$\sigma =\frac{L}{RS},$$
where *σ* is the material conductivity, Ω·m; *R* is the material resistance, Ω; *S* is the cross-sectional area of the material, m^2^; and *L* is the length of the material., m.

From Formula (4), it can be observed that conductivity is inversely proportional to resistance. In this study, due to the constant electrode spacing and driving voltage, conductivity can be characterized via measuring resistance. Subsequently, five seed cotton samples (150 g per sample) with an MR of 9.37% were used. A sample box with a diameter of 16 cm was selected, and the experiment was conducted at 25 °C. Each seed cotton sample was compressed from 90~250 kg·m^−3^, while reducing the compression height from 10 cm to 3.5 cm. Each sample was compressed sixty-six times, with each compression stroke being 1 mm. The contact pressure, resistance value, and compression density during the experiment were recorded. Based on the recorded data, a graph illustrating the relationships between contact pressure, resistance value, and compression density was obtained.

#### 2.3.4. The MR Measurement Model

In this experiment, thirty-two different MR levels were selected, with two samples for each level. A sample box with a diameter of 16 cm was used. Based on the parameters of the John Deere 7760 cotton picker [28], the packing density was calculated to be 186.9~207.7 kg·m^−3^. Therefore, the compression density range was set to 150~250 kg·m^−3^, and the compression height was reduced from 5 cm to 3 cm. Each sample underwent eleven loadings, with a downward displacement of 2 mm per time. During the experiment, contact pressure, resistance value, and MR data were recorded, so a total of seven hundred and four data were obtained (32 MR levels × 2 samples × 11 densities = 704). The contact pressure and resistance value were chosen as the feature variables for predicting the seed cotton MR. After normalizing, the dataset was split into training, testing, and validation datasets in an 8:1:1 ratio. Then, support vector regression (SVR), random forest (RF), and a back propagation neural network (BPNN) were employed to build the models. Cross-validation was used to optimize the model parameters. The performance of the models was evaluated using the coefficient of determination (R^2^) and root mean square error (RMSE). To further investigate the feasibility of the resistive sensing of MR method based on pressure compensation, the built models were validated using data from the validation set (71). Finally, a performance evaluation test was conducted on the method, using RMSE and the coefficient of variation (CV) as evaluation indicators.

### 2.4. Data Processing

The experimental data were recorded using Excel software (Version 2016, Microsoft Corporation, Redmond, WA, USA). The data were analyzed using SPSS Statistics software (Version 26, IBM Corporation, Armonk, NY, USA), and the regression analysis of compression density, contact pressure, and resistance was conducted using both linear and nonlinear functions (allometric function and exp3p2 function). The R^2^ and RMSE were used to assess the goodness of fit. Furthermore, the graphs were drawn using Origin software (Version 2021, OriginLab Corporation, Northampton, MA, USA).

## 3. Results and Discussion

### 3.1. Equipment Calibration

To ensure the reliability of the experiment results, this study employed standard resistors and weights to calibrate the LCR bridge tester and pressure sensor, respectively. The Bland–Altman plot was used to analyze the measurement errors (differences between the measurement values and the standard values). To eliminate the influence of different orders of magnitude and provide an intuitive representation of the error magnitude and direction, the error results were presented in the form of difference (%) (Figure 4).

When calibrating using twenty standard resistors with different resistance, the measurement results were all within the 95% confidence interval. Additionally, most of the samples showed small errors, and measured values were close to the mean value (Figure 4a). When calibrating using twenty-one weights with different weights, the measurement results were mostly within the 95% confidence interval. Similarly, the errors were small, and measured values were close to the mean value (Figure 4b). However, when calibrating using small weights, there was a noticeable deviation. This deviation was caused by the measurement accuracy and range of the pressure sensor.

### 3.2. Mechanical Analysis of Seed Cotton Compression

Compared to other agricultural materials, seed cotton exhibited more complex mechanical changes during the compression process due to the entanglement of cotton fibers on the cottonseeds. Five seed cotton samples were compressed at a speed of 10 mm·s^−1^. It was found that the pressure–time curves obtained exhibit similar characteristics with small difference (Figure 5a).

During 0~6 s, the seed cotton was compressed from its natural state (Figure 5b). The spaces between cotton fibers rapidly decreased, and there was a significant buffer space within the cotton fiber aggregate. At this stage, the contact pressure was relatively low, with a small variation, exhibiting an approximate linear relationship with strain. This stage was defined as the compression stage. During 6~12 s, the buffer space within the cotton fiber aggregate decreased. Due to the random arrangement of the cottonseeds, some slippage occurred during the compression. As strain increased, the increase in contact pressure became gentle. This stage was defined as the transition stage. During 12~16 s, the cotton fiber aggregate had a minimal buffer space and a tightly packed internal arrangement. The seed cotton underwent elastic deformation. At this time, small strains resulted in a rapid increase in contact pressure, leading to significant changes. The relationship between contact pressure and strain was approximately linear in this stage. This stage was defined as the densification stage. Under alternating stress, due to the strain lagging behind stress, when loading was stopped (16~61 s), the internal strain within the cotton fiber aggregate gradually decreased. Consequently, the contact pressure decreased nonlinearly, initially at a faster rate and then gradually stabilizing. This stage was defined as the relaxation stage.

### 3.3. Analysis of Influencing Factors of Contact Pressure

This experiment investigated the factors influencing contact pressure by manipulating the compression diameter, compression height, and compression density to simulate the changes in cotton bale parameters during the densification.

According to Figure 6a, when the compression height and density were constant, the contact pressure remained within a stable range, with insignificant variations. This indicates that the compression diameter does not significantly affect the statistical analysis conclusions of the dataset. Similarly, according to Figure 6b, when the compression height and density were constant, the contact pressure of the seed cotton also remained within a stable range, with insignificant variations. This indicates that the compression height does not impact the statistical analysis conclusions of the dataset. However, according to Figure 6c, when the compression height and density were constant, the contact pressure increased significantly with the increase in density. Moreover, as the density increased, the dispersion of the base pressure also increased. This implies that the compression density has a significant impact on the statistical analysis conclusions of the dataset. Therefore, based on Figure 6, it can be deduced that during the packaging process of seed cotton, there is a weak correlation between contact pressure and compression volume, while there is a strong correlation between contact pressure and compression density. The cotton bale density can be indirectly manifested through the contact pressure.

### 3.4. Analysis of Influencing Factors of Conductivity

The analysis results of the correlations among compression density, contact pressure, and resistance values (Figure 7) showed that there was a high positive correlation (0.91) between pressure and density. There was also a negative correlation between density and resistance (−0.8) and between pressure and resistance (−0.59). Furthermore, the *p*-values for the significance tests of all factors were less than 0.01. Thus, the correlations between the parameters were statistically significant, indicating that further exploration could be conducted.

The linear and nonlinear regression analyses (allometric function and exp3p2 function) of contact pressure and density during the densification showed (Figure 8) that as the density increased, the pressure exhibited a nonlinear increase. Additionally, with the continuous increase in density, the contact pressure tended to be scattered. This may be due to the fact that after completing the measurement of a sample, the next cotton sample needs to be repositioned. The arrangement of cotton fibers has a certain degree of randomness, and an uneven placement of the cotton sample can result in uneven density distribution in the cotton fiber aggregate, leading to different contact pressures at the pressure sensor. After entering the transition and densification stages, the compressibility of the seed cotton gradually reduces, and the contact pressure changes greatly with the increase in density. The fitting results are shown in Table 2. Comparing the R^2^ and RMSE values of the three models, it was found that the nonlinear function models provided a better fit than the linear function models. Besides, among the nonlinear models, the Exp3P2 function model had the highest R^2^ and the lowest RMSE, indicating the best fit. Therefore, $y=e^{\left(a+bx+cx^{2}\right)}$ is the optimal analytical equation for characterizing the relationship between the density of the cotton fiber aggregate and the pressure.

The regression analysis of resistance and density during the densification process using the linear and nonlinear functions (allometric function and exp3p2 function) found that as the density increased, the corresponding resistance decreased (Figure 9). This may be due to the fact that during the compression of the cotton fiber aggregate, the density gradually increases, the spaces between fibers reduces, and the number of contact points increases. This significantly increases the parallel current paths between the cotton fibers, leading to a decrease in resistance. Additionally, it was also found that when the density was approximately 80~90 kg·m^−3^, the resistance decreased rapidly with the increase in density. As the cotton fiber aggregate continued to be compressed, when the density was around 90~140 kg·m^−3^, the resistance gradually decreased with the increase in density. When the density exceeded 160 kg·m^−3^, the resistance decreased slowly with the increase in density, approaching a certain limit. The fitting results are shown in Table 3. Comparing the R^2^ and RMSE values of the three models, it was found that the nonlinear function models provided a better fit than the linear function models. Moreover, among the nonlinear models, the exp3P2 function model had the highest R^2^ and the lowest RMSE, indicating the best fit. Therefore, $y=e^{\left(a+bx+cx^{2}\right)}$ is the optimal analytical equation for characterizing the relationship between the density of the cotton fiber aggregate and the pressure.

The regression analysis of contact pressure and resistance using both linear and nonlinear functions (allometric function and exp3p2 function) found that as the contact pressure increased, the corresponding resistance decreased (Figure 10). This may be due to the fact that high contact pressure leads to a high density, which further results in strong conductivity. Additionally, it was also found that when the contact pressure was less than 0.1 kg, the resistance decreased rapidly with the increase in contact pressure. When the contact pressure was in the range of 0.1~0.5 kg, the resistance gradually decreased with the increase in contact pressure. When the contact pressure exceeded 0.5 kg, the resistance decreased slowly with the increase in density, approaching a certain limit. The fitting results are shown in Table 4. Comparing the R^2^ and RMSE values of the three models, it was found that the nonlinear function models provided a better fit than the linear function models. Moreover, among the nonlinear models, the allometric function model had the highest R^2^ and the lowest RMSE, indicating the best fit. Therefore, $y=ax^{b}$ is the optimal analytical equation for characterizing the relationship between the density of cotton fiber aggregate and the contact pressure.

### 3.5. Model Performance Comparison and Validation

It can be seen from the above results that there is a significant nonlinear correlation between density and contact pressure, as well as between density and resistance value. By measuring the contact pressure, the compression density can be indirectly reflected. Therefore, the contact pressure and resistance can be used as feature variables in the MR prediction model to investigate the relationships between the resistance, contact pressure, and MR. Seed cotton with different MR showed significant differences in resistance under different contact pressures (Figure 11). With a constant MR, the resistance showed a nonlinear decrease as the contact pressure increased. Similarly, with a constant contact pressure, the resistance also showed a nonlinear decrease as the MR increased; that is, the contact pressure affects the resistance measurement of seed cotton, which, in turn, affects the measurement of MR. Therefore, using contact pressure as a compensation value in the resistance-based MR measurement can improve the accuracy of the measurement.

To achieve MR measurement during the compression process of seed cotton during the harvesting operations, it is necessary to build an MR prediction model based on the experimental results above. In this study, contact pressure and resistance were chosen as the features for the prediction model. Three algorithms, namely, SVR, RF, and BPNN, were utilized for modeling. The prediction accuracy of each model was evaluated using the testing dataset (Figure 12).

To explore the optimal seed cotton MR prediction model, R^2^ and RMSE were used to evaluate the performance of the built models, and then the prediction performances of the three models were compared. It was found that the R^2^ and RMSE of SVR model were 0.974 and 0.296%, respectively; the R^2^ and RMSE of RF model were 0.977 and 0.261%, respectively; and those of BPNN model were 0.986 and 0.204%, respectively (Table 5).

It can be seen that the predicted MR of BPNN model was the closest to the actual MR among the three models (Table 5). BPNN is a type of three-layer feedforward structure. The three layers are the input layer, hidden layer, and output layer (Figure 13). The input layer receives information (resistance values and pressure values) from external sources and passes it on to the network for processing. The hidden layer receives information from the input layer and processes all the information. The output layer receives the processed information from the network and sends the resulting outputs to external receptors. The input signals are modified via interconnected weights known as weight factors *v_mn_*, which represent the interconnections from the *m*th node of the first layer to the *n*th node of the second layer. These modified signals are then adjusted using the tanh transfer function to compute the total activation. Similarly, the output signals from the hidden layer are adjusted by the interconnection weights *w_ij_* from the *i*th node of the output layer to the *j*th node of the hidden layer. The adjusted signals are summed using the tanh transfer function, and the resulting outputs are collected at the output layer.

BPNN has strong robustness, memory ability, nonlinear mapping ability, and self-learning ability. Therefore, it has good applicability to the experimental data in this study. BPNN was used to build the seed cotton MR prediction model, and the validation dataset was used to test the stability of the prediction model. It was found that when the BPNN model was used to predict the MR of each sample, all errors were small (Figure 14). Specifically, the calculation of the standard deviation for each dataset showed that the mean absolute error was less than 0.26%. This indicates the high stability of the model.

### 3.6. Performance Evaluation

To verify the reliability of the MR measurement method based on pressure compensation, an additional set, including six groups of seed cotton samples with different MR levels, was prepared. Each group included two samples. One sample was measured using the experimental platform, and the MR value was displayed on the computer (Figure 15). The other sample was measured using a semi-automatic constant-temperature oven. The results from both methods were compared.

It was found that the proposed MR measurement method exhibited a relatively high measurement accuracy, with the maximum RMSE and mean RMSE between the two methods being 0.24% and 0.20%, respectively (Table 6). This indicates that the pressure compensation-based method for measuring the seed cotton MR is reliable. Furthermore, the maximum CV and mean CV of the measured values on the experimental platform were 3.66% and 2.22%, respectively. This demonstrates the relatively high stability for seed cotton samples with different MR levels. Based on these findings, it can be concluded that even in the case of density changes, the method for resistive sensing of seed cotton MR measurement based on pressure compensation still has relatively high accuracy and stability, so it can be applied in practical measurements.

## 4. Conclusions

By simulating the changes in cotton bale parameters during the packing process of a cotton picker, it was found that there was a weak correlation between contact pressure and compression volume, while a significant correlation was observed between contact pressure and compression density. Therefore, the contact pressure can be used to characterize the compression density. In addition, nonlinear correlations between compression density, contact pressure, and conductivity were found. The prediction accuracy of the nonlinear models was significantly higher than that of linear models. Thus, contact pressure can be used as a compensatory value in the resistive sensing of seed cotton MR. Finally, a new method for the resistive sensing of seed cotton MR measurement based on pressure compensation was proposed. An MR experimental platform was designed, and an MR prediction model was built using experimental data. The BPNN model had the best prediction accuracy, with an R^2^ of 0.986 and an RMSE of 0.204%. The performance of the experimental platform equipped with the BPNN model was tested, and we found that the mean RMSE was 0.20%, and the mean CV was 2.22%.

To sum up, the pressure compensation-based resistive sensing for measuring MR in seed cotton proposed in this study has relatively high accuracy and reliability. The research findings can provide a technical reference for online measurement of MR during cotton harvesting operations. This method can also be applied in the online measurement of MR in raw cotton and the moisture content measurement in grains and forages.

## Figures and Tables

**Figure 1 sensors-23-08421-f001:**
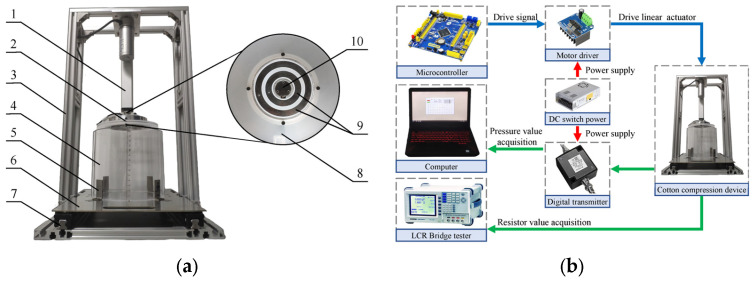
Experimental platform: (**a**) cotton compressing device (1. DC linear actuator; 2. Seed cotton MR sensor; 3. Aluminum frame; 4. Sample box; 5. Sample box holder; 6. Plate; 7. Slide rail; 8. Cotton compressing plate; 9. Sensing electrodes; 10. Pressure sensor.); (**b**) hardware connection.

**Figure 2 sensors-23-08421-f002:**
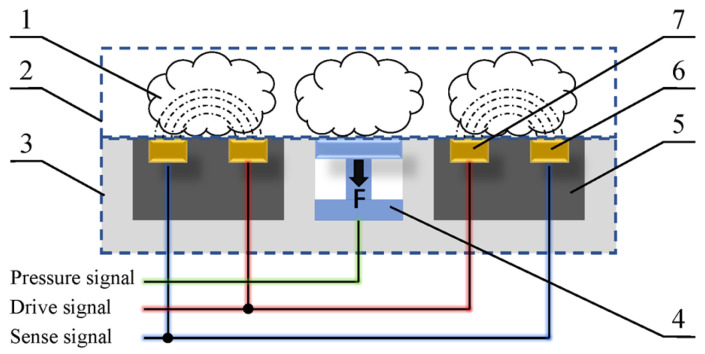
Structure of seed cotton moisture regain sensor: 1. Seed cotton; 2. Cotton fiber aggregate; 3. Probe; 4. Pressure sensor; 5. Insulating material; 6. Sense electrode; 7. Drive electrode.

**Figure 3 sensors-23-08421-f003:**
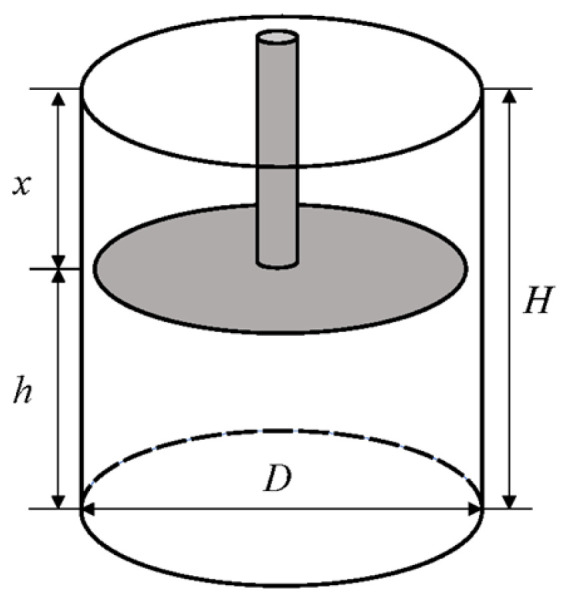
Structure of the cotton compressing device.

**Figure 4 sensors-23-08421-f004:**
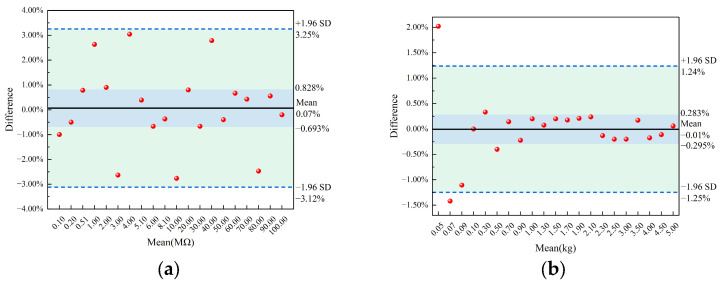
Mechanical analysis diagram of compression of seed cotton: (**a**) resistance value calibration result; (**b**) pressure value calibration result.

**Figure 5 sensors-23-08421-f005:**
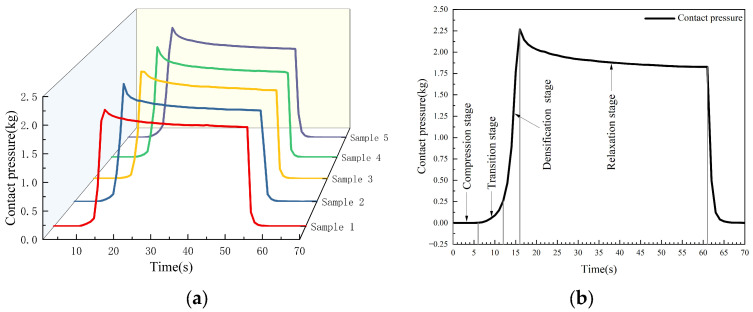
Mechanical analysis of seed cotton: (**a**) pressure–time curve during seed cotton compression; (**b**) stage division in seed cotton compression process.

**Figure 6 sensors-23-08421-f006:**
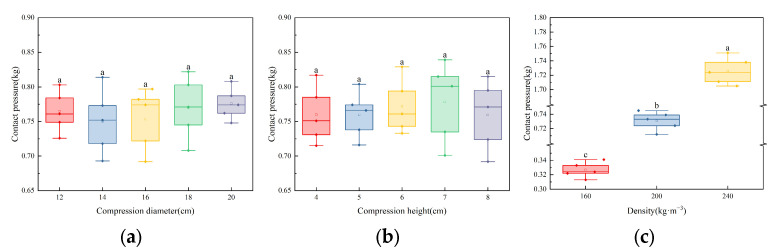
Influence of diameter, height, and density on contact pressure: (**a**) same density and height; (**b**) same density and diameter; (**c**) same height and diameter.

**Figure 7 sensors-23-08421-f007:**
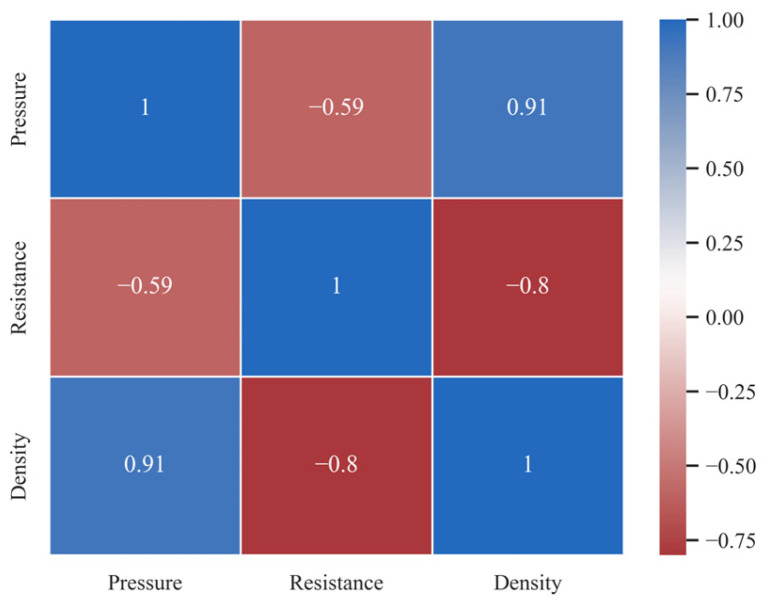
Correlation confusion matrix.

**Figure 8 sensors-23-08421-f008:**
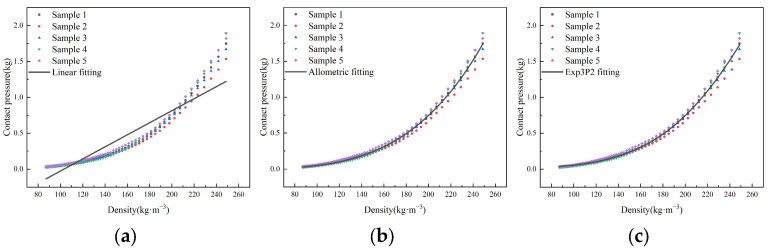
Regression curves of density and contact pressure during compression: (**a**) linear function; (**b**) allometric function; (**c**) exp3P2 function.

**Figure 9 sensors-23-08421-f009:**
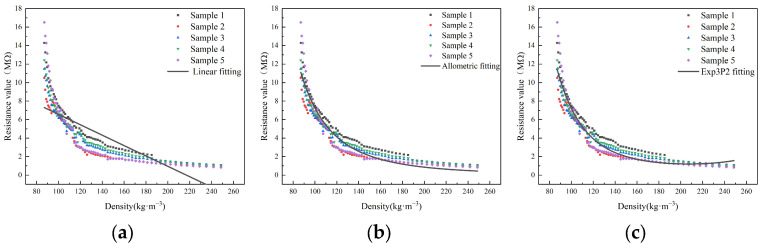
Regression curves of density and resistance during compression: (**a**) linear function; (**b**) allometric function; (**c**) exp3P2 function.

**Figure 10 sensors-23-08421-f010:**
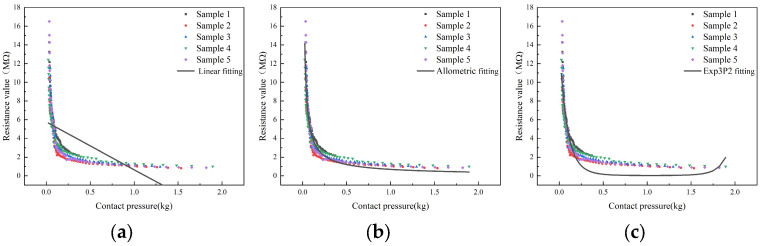
Regression curves of contact pressure and resistance during compression: (**a**) linear function; (**b**) allometric function; (**c**) exp3P2 function.

**Figure 11 sensors-23-08421-f011:**
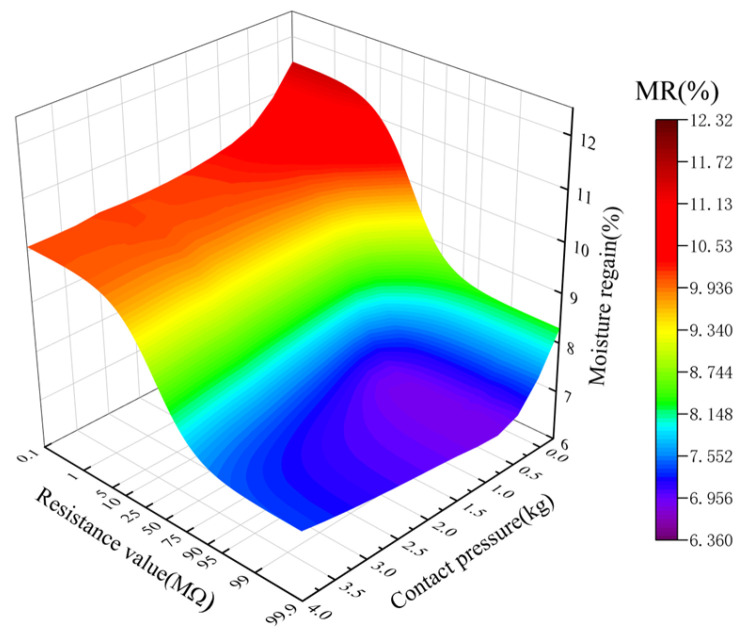
Correlations between resistance, moisture regain (MR), and contact pressure.

**Figure 12 sensors-23-08421-f012:**
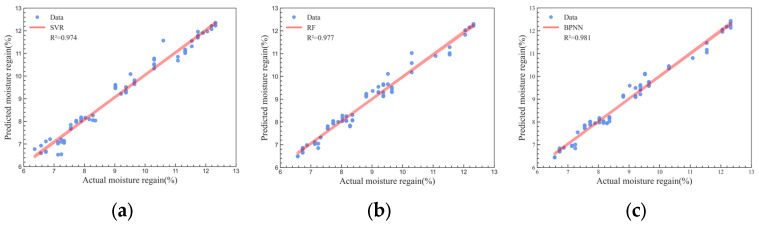
Evaluation of prediction accuracy for different models: (**a**) SVR; (**b**) RF; (**c**) BPNN.

**Figure 13 sensors-23-08421-f013:**
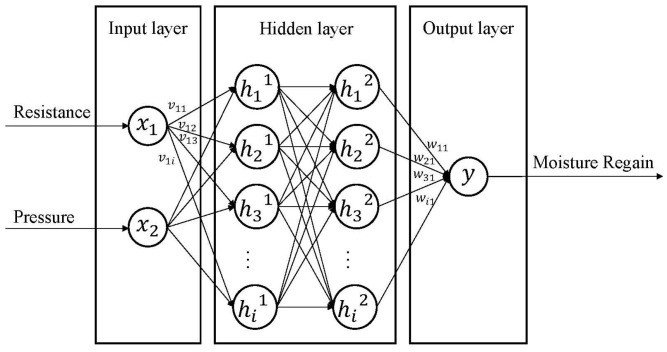
BPNN architecture diagram of seed cotton moisture regain detection model.

**Figure 14 sensors-23-08421-f014:**
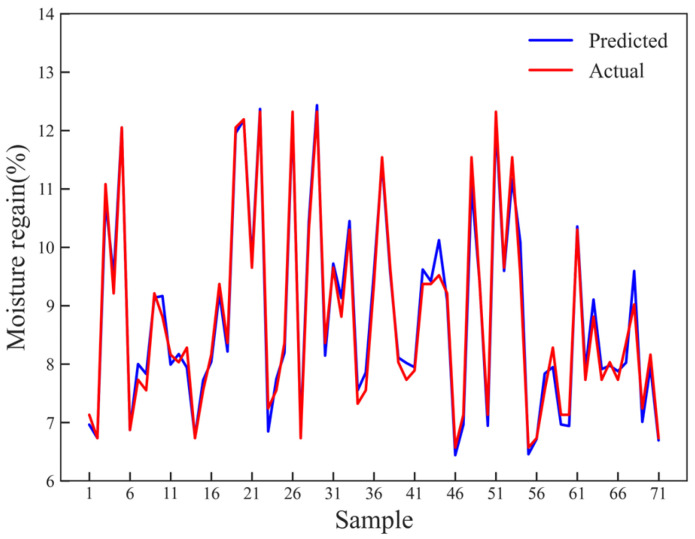
Comparison between the model-predicted values and the actual values.

**Figure 15 sensors-23-08421-f015:**
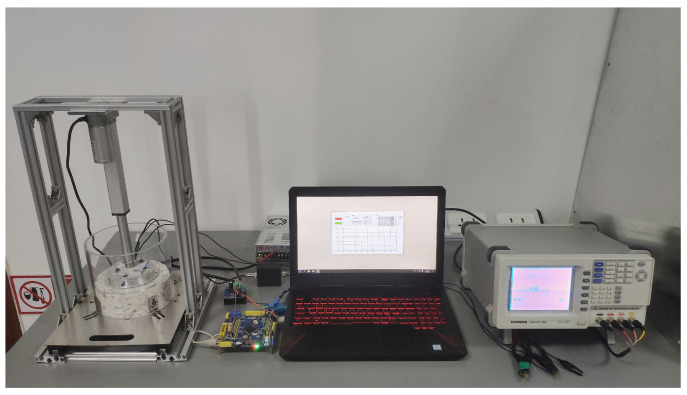
Performance evaluation test.

**Table 1 sensors-23-08421-t001:** The cotton samples’ MR at different temperatures and relative humidities.

Relative Humidity (%)	Temperature (°C)							
	5	10	15	20	25	30	35	40
60	7.38%	7.29%	7.18%	7.06%	6.92%	6.78%	6.62%	6.41%
70	8.44%	8.36%	8.26%	8.14%	8.00%	7.84%	7.66%	7.42%
80	10.33%	10.18%	10.01%	9.82%	9.61%	9.38%	9.13%	8.85%
90	12.68%	12.57%	12.37%	12.14%	11.88%	11.6%	11.31%	11.03%

**Table 2 sensors-23-08421-t002:** Regression models of contact pressure and density.

Category	Expression	Model Parameter			R^2^	RMSE
		a	b	c		
Linear function	*y = a + bx*	−8.660 × 10^−1^	8.390 × 10^−3^	-	0.865	0.147
Allometric function	*y = ax^b^*	7.508 × 10^−10^	3.909	-	0.993	0.033
Exp3P2 function	$y=e^{\left(a+bx+cx^{2}\right)}$	−6.618	4.300 × 10^−2^	−5.759 × 10^−5^	0.993	0.032

**Table 3 sensors-23-08421-t003:** Regression models of resistance and density.

Category	Expression	Model Parameter			R^2^	RMSE
		a	b	c		
Linear function	*y = a + bx*	5.617 × 10^−2^	1.218 × 10	-	0.641	1.869
Allometric function	*y = ax^b^*	1.004 × 10^−7^	−3.071	-	0.921	0.875
Exp3P2 function	$y=e^{\left(a+bx+cx^{2}\right)}$	6.846	−6.400 × 10^−2^	1.547 × 10^−4^	0.929	0.830

**Table 4 sensors-23-08421-t004:** Regression models of resistance and contact pressure.

Category	Expression	Model Parameter			R^2^	RMSE
		a	b	c		
Linear function	*y = a + bx*	5.761	−5.110	-	0.354	2.508
Allometric function	*y = ax^b^*	6.960 × 10^−1^	−8.070 × 10^−1^	-	0.893	1.021
Exp3P2 function	$y=e^{\left(a+bx+cx^{2}\right)}$	2.669	−1.171 × 10	5.627	0.829	1.291

**Table 5 sensors-23-08421-t005:** Comparison of performance evaluation metrics of different predictive models.

Algorithm Model	Model Performance Evaluation Indicators	
	R^2^	RMSE
SVR	0.974	0.296%
RF	0.977	0.261%
BPNN	0.986	0.204%

**Table 6 sensors-23-08421-t006:** MR measurement results.

Group	Oven Method (%)	Experimental Platform (%) *												RMSE (%)	CV (%)
		*ρ* _1_	*ρ* _2_	*ρ* _3_	*ρ* _4_	*ρ* _5_	*ρ* _6_	*ρ* _7_	*ρ* _8_	*ρ* _9_	*ρ* _10_	*ρ* _11_	Mean		
1	5.94	5.94	5.95	5.85	5.74	5.81	5.77	6.02	6.36	6.20	6.03	6.42	6.01	0.23	3.66
2	6.88	6.54	6.82	6.68	6.76	6.63	7.05	7.18	7.09	7.20	7.15	7.16	6.93	0.24	3.43
3	7.96	7.95	7.88	8.13	8.11	8.08	8.14	7.94	7.93	7.73	8.02	8.08	8.00	0.13	1.51
4	9.39	9.21	9.16	9.26	9.52	9.66	9.72	9.69	9.54	9.64	9.39	9.45	9.48	0.21	2.01
5	10.51	10.40	10.47	10.46	10.42	10.28	10.26	10.39	10.55	10.73	10.83	10.86	10.51	0.20	1.88
6	11.77	11.86	11.80	11.78	11.67	11.59	11.51	11.65	11.68	11.63	11.57	11.72	11.68	0.23	0.86
Mean														0.20	2.22

* Measurement of MR of the experimental platform under different densities: *ρ*_1_ = 149 kg·m^−3^; *ρ*_2_ = 155 kg·m^−3^; *ρ*_3_ = 162 kg·m^−3^; *ρ*_4_ = 170 kg·m^−3^; *ρ*_5_ = 178 kg·m^−3^; *ρ*_6_ = 187 kg·m^−3^; *ρ*_7_ = 196 kg·m^−3^; *ρ*_8_ = 207 kg·m^−3^; *ρ*_9_ = 219 kg·m^−3^; *ρ*_10_ = 233 kg·m^−3^; *ρ*_11_ = 249 kg·m^−3^.

## Data Availability

The data presented in this study are available on request from the corresponding author.

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
