# Peer review of "Resistive Sensing of Seed Cotton Moisture Regain Based on Pressure Compensation"

_sensors, 2023, doi:10.3390/s23208421_

Round 1
Reviewer 1 Report
The manuscript entitled “Resistive Sensing of Seed Cotton Moisture Regain Based on Pressure Compensation” proposed to method for evaluating the Moisture Regain (MR) of cotton seeds. The work in its entirety is well structured and written clearly and concisely.
The abstract clearly describes the problem and the main findings of the study; in addition to concluding by showing the applicability of these findings. Very good!
The introduction, methodology, results, discussion, and conclusion are very well described.
The English of the manuscript is good!
The only negative point of the manuscript is the low number of citations of recent articles (last 3 years). However, this aspect is understandable given the theme. More importantly, it is a research paper with an important technological strategy.
Finally, I consider the manuscript suitable and recommend that it be published in Sensors.
Author Response
Response to Reviewer 1 Comments
Title: Resistive Sensing of Seed Cotton Moisture Regain Based on Pressure Compensation
Manuscript ID: sensors-2606782
Authors: Liang Fang, Ruoyu Zhang, Hongwei Duan, Jinqiang Chang, Zhaoquan Zeng, Yifu Qian, Mianzhe Hong
This is the authors’ reply to the comments and suggestions made by the reviewer 1 of the original manuscript entitled “Resistive Sensing of Seed Cotton Moisture Regain Based on Pressure Compensation”. First of all, we would like to thank the reviewer 1 for volunteering his/her valuable time in handling the review of our submission. We highly appreciate the reviewer 1’ comments and constructive remarks. This submission has been carefully revised based on the suggestions made by reviewer 1, which major changes are highlighted in red. We hope reviewer 1 will be satisfied with the revised manuscript. In the following, please kindly find the point-to-point responses.
Reviewer 1: “(1) The manuscript entitled “Resistive Sensing of Seed Cotton Moisture Regain Based on Pressure Compensation” proposed to method for evaluating the Moisture Regain (MR) of cotton seeds. The work in its entirety is well structured and written clearly and concisely.”
Authors’ Reply: We would like to express our gratitude to the reviewer 1 for his/her positive feedback on our manuscript. We are delighted to hear that the structure and clarity of the manuscript met his/her expectations. We sincerely appreciate the reviewer 1's valuable comments.
Reviewer 1: “(2) The abstract clearly describes the problem and the main findings of the study; in addition to concluding by showing the applicability of these findings. Very good!”
Authors’ Reply: We would like to express our gratitude to the reviewer 1 for his/her positive feedback on our manuscript. We are glad that the abstract effectively conveys the problem addressed, the main findings, and highlights the practical implications of our research. We appreciate the reviewer's acknowledgement of the clarity and coherence of the abstract. Their encouraging remarks motivate us to further enhance the manuscript. We sincerely appreciate the reviewer 1's valuable comments.
Reviewer 1: “(3) The introduction, methodology, results, discussion, and conclusion are very well described.”
Authors’ Reply: We sincerely appreciate the reviewer 1's positive feedback regarding the description of the introduction, methodology, results, discussion, and conclusion in our manuscript. We put significant effort into ensuring clarity and coherence throughout these sections, and the reviewer's acknowledgment reinforces our confidence in his/her effectiveness. We sincerely appreciate the reviewer 1's valuable comments.
Reviewer 1: “(4) The English of the manuscript is good!”
Authors’ Reply: We would like to express our appreciation to the reviewer 1 for his/her positive feedback on the English of our manuscript. We strive to ensure that the language used in the manuscript is clear, concise, and grammatically correct. Reviewer 1's feedback serves as an encouragement to our team. We acknowledge the importance of effective communication in scientific writing and are grateful for reviewer 1's recognition of our efforts. We sincerely appreciate the reviewer 1's valuable comments.
Reviewer 1: “(5) The only negative point of the manuscript is the low number of citations of recent articles (last 3 years). However, this aspect is understandable given the theme. More importantly, it is a research paper with an important technological strategy.”
Authors’ Reply: We would like to express our gratitude to the reviewer 1 for his/her positive feedback on our manuscript. We understand and acknowledge the concern regarding the low number of citations of recent articles, particularly within the last three years. The nature of the theme addressed in our research may limit the availability of recent references. However, we have made every effort to include relevant and important references in the original manuscript to support our work. Furthermore, we greatly appreciate reviewer 1's recognition of the importance of the technological strategy presented in our research paper.
We thank reviewer 1 again for his/her valuable suggestions and comments that have helped us improve the quality of this manuscript. We aimed to provide innovative and valuable insights in our study, and reviewer 1's acknowledgment encourages us to continue exploring new avenues in this field. We hope reviewer 1 will be satisfied with our clarification and modification.

Reviewer 2 Report
Moisture content on-line detection is difficult for stored products like cotton. This study is interesting and a new method for seed cotton moisture was developed. Now the principle and details should be supplemented.
(1) The equilibrium moisture content in each sample at different combination of temperature and relative humidity should be given. The accuracy of the constant temperature and humidity test chamber should be given.
(2) How to determine the density of cotton? The more formula are needed. The difference between cotton density and moisture content in the present study and the previous reported values should be given.
(3) The resistance value of cotton is used for determination of moisture content. A method for determination of the resistance values of cotton in figure 11 were should be given in the materials and methods. The relationship between conductivity and resistance value should be introduced.
(4) The authors used the compressed volume, compression density, contact pressure and conductivity of seed cotton to show the cotton moisture content. The principle should be given.
(5) The difference between a moisture determination method and oven method for stored products is below 0.5%. In table 6, the resistance method for cotton moisture content should be given. It is better to give the means and standard errors in table 6.
Author Response
Response to Reviewer 2 Comments
Title: Resistive Sensing of Seed Cotton Moisture Regain Based on Pressure Compensation
Manuscript ID: sensors-2606782
Authors: Liang Fang, Ruoyu Zhang, Hongwei Duan, Jinqiang Chang, Zhaoquan Zeng, Yifu Qian, Mianzhe Hong
This is the authors’ reply to the comments and suggestions made by the reviewer 2 of the original manuscript entitled “Resistive Sensing of Seed Cotton Moisture Regain Based on Pressure Compensation”. First of all, we would like to thank the reviewer 2 for volunteering his/her valuable time in handling the review of our submission. We highly appreciate the reviewer 2’ comments and constructive remarks. This submission has been carefully revised based on the suggestions made by reviewer 2, which major changes are highlighted in red. We hope reviewer 2 will be satisfied with the revised manuscript. In the following, please kindly find the point-to-point responses.
Reviewer 2: “(1) The equilibrium moisture content in each sample at different combination of temperature and relative humidity should be given. The accuracy of the constant temperature and humidity test chamber should be given.”
Authors’ Reply: We thank reviewer 2 for his/her valuable comments and suggestions on the manuscript. According to reviewer 2’s comments, in the revised manuscript, we have provided the equilibrium moisture regain of each sample under different temperature and relative humidity combinations in the revised manuscript [line 116]. This information will further support our research findings and provide a more comprehensive analysis of the study. In addition, we will also provide information on the accuracy of the constant temperature and humidity test chamber used in the experiment [line 101 to 106].
The revised manuscript [line 116] now reads as follows:
Table 1. The cotton samples MR at different temperatures and relative humidities.
Relative humidity (%) |
Temperature (℃) |
|||||||
5 |
10 |
15 |
20 |
25 |
30 |
35 |
40 |
|
60 |
7.38% |
7.29% |
7.18% |
7.06% |
6.92% |
6.78% |
6.62% |
6.41% |
70 |
8.44% |
8.36% |
8.26% |
8.14% |
8.00% |
7.84% |
7.66% |
7.42% |
80 |
10.33% |
10.18% |
10.01% |
9.82% |
9.61% |
9.38% |
9.13% |
8.85% |
90 |
12.68% |
12.57% |
12.37% |
12.14% |
11.88% |
11.6% |
11.31% |
11.03% |
The revised manuscript [line 101 to 106] now reads as follows:
Secondly, the cotton samples were placed in the constant temperature and humidity test chamber (FQY/WSK-400C, Guangzhou Fengqianyuan Environmental Test Equipment Co., Ltd.), with a temperature deviation of ±2.0 ℃ and a humidity deviation of ±3.0 %RH. And balance for 24 hours at different temperatures and relative humidity levels, to prepare cotton samples with different MR levels (Table 1).
We hope reviewer 2 will be satisfied with the revised manuscript.
Reviewer 2: “(2) How to determine the density of cotton? The more formulas are needed. The difference between cotton density and moisture content in the present study and the previous reported values should be given.”
Authors’ Reply: We thank reviewer 2 for his/her valuable comments and suggestions on the manuscript. However, the calculation method for cotton density has been provided in Equation (3) in original manuscript. Cotton density is only used to explore the relationship between density and pressure, demonstrating that density can be characterized by pressure. The corresponding equation was given in Table 2. In the subsequent measurement of cotton moisture regain, only pressure and resistance values were used, and density was not involved. On the other hand, according to reviewer 2’s comments, in the revised manuscript, we have compared the method of measuring cotton moisture regain in our study with previous reports [line 76 to 79]. This will help provide a context for understanding the novelty and significance of our findings.
The revised manuscript [line 76 to 79] now reads as follows:
In summary, there is limited research on the measurement of seed cotton MR during the harvesting operations, and most studies have not considered the impact of changes in cotton density on measurement results. This greatly affects the measurement accuracy.
We hope reviewer 2 will be satisfied with the revised manuscript.
Reviewer 2: “(3) The resistance value of cotton is used for determination of moisture content. A method for determination of the resistance values of cotton in figure 11 were should be given in the materials and methods. The relationship between conductivity and resistance value should be introduced.”
Authors’ Reply: We thank reviewer 2 for his/her valuable comments and suggestions on the manuscript. However, in this study, the resistive values of seed cotton were measured using LCR bridge tester, and the method for measuring the resistive values of cotton was provided in the Materials and Methods section of the original manuscript [line 143 to 145]. On the other hand, according to reviewer 2’s comments, in the revised manuscript, we have introduced the relationship between conductivity and resistance values in the Materials and Methods section. This will help readers understand the basic principle and significance of using electrical resistance values to measure moisture content [line 184 to 190].
The revised manuscript [line 184 to 190] now reads as follows:
The relationship between conductivity and resistance was analyzed, and the calculation formula between the two is as follows:
|
(4) |
Where σ is the material conductivity, Ω·m; R is the material resistance, Ω; S is the cross-sectional area of the material, m2; L is the length of the material., m.
From the formula (4), it can be observed that conductivity is inversely proportional to resistance. In this study, due to the constant electrode spacing and driving voltage, conductivity can be characterized by measuring resistance.
We hope reviewer 2 will be satisfied with the revised manuscript.
Reviewer 2: “(4) The authors used the compressed volume, compression density, contact pressure and conductivity of seed cotton to show the cotton moisture content. The principle should be given.”
Authors’ Reply: We thank reviewer 2 for his/her valuable comments and suggestions on the manuscript. However, we have provided the principle of resistive sensing of seed cotton MR based on pressure compensation [line 128 to 139]. On the other hand, according to reviewer 2’s comments, in the revised manuscript, we have added the architecture diagram of BPNN to explain the principle of moisture regain measurement more clearly at the algorithmic level [line 390 to 404].
The revised manuscript [line 390 to 404] now reads as follows:
BPNN is a type of three-layer feedforward structure. The three layers are the input layer, hidden layer, and output layer (Figure 13). The input layer receives information (resistance values and pressure values) from external sources and passes it to the network for processing. The hidden layer receives information from the input layer and processes all the information. The output layer receives the processed information from the network and sends the resulting outputs to external receptors. The input signals are modified by interconnected weights known as weight factors vmn, which represent the interconnections from the mth node of the first layer to the nth node of the second layer. These modified signals are then adjusted using the tanh transfer function to compute the total activation. Similarly, the output signals from the hidden layer are adjusted by the interconnection weights wij from the ith node of the output layer to the jth node of the hidden layer. The adjusted signals are summed using the tanh transfer function, and the resulting outputs are collected at the output layer.
Figure 13. BPNN architecture diagram of seed cotton moisture regain detection model
We hope reviewer 2 will be satisfied with the revised manuscript.
Reviewer 2: “(5) The difference between a moisture determination method and oven method for stored products are below 0.5%. In table 6, the resistance method for cotton moisture content should be given. It is better to give the means and standard errors in table 6.”
Authors’ Reply: We thank reviewer 2 for his/her valuable comments and suggestions on the manuscript. However, the testing platform in Table 6 has been introduced in the Materials and Methods section, which is based on the construction of seed cotton moisture regain resistive sensing method using pressure compensation [line 118 to 124]. On the other hand, according to reviewer 2’s comments, in the revised manuscript, we have provided the means and standard errors (Root mean squared error, RMSE) of the data in Table 6.
The revised manuscript [line 423] now reads as follows:
Table 6. MR measurement results.
Group |
Oven method (%) |
Experimental platform (%) * |
RMSE (%) |
CV (%) |
|||||||||||
ρ1 |
ρ2 |
ρ3 |
ρ4 |
ρ5 |
ρ6 |
ρ7 |
ρ8 |
ρ9 |
ρ10 |
ρ11 |
Mean |
||||
1 |
5.94 |
5.94 |
5.95 |
5.85 |
5.74 |
5.81 |
5.77 |
6.02 |
6.36 |
6.20 |
6.03 |
6.42 |
6.01 |
0.23 |
3.66 |
2 |
6.88 |
6.54 |
6.82 |
6.68 |
6.76 |
6.63 |
7.05 |
7.18 |
7.09 |
7.20 |
7.15 |
7.16 |
6.93 |
0.24 |
3.43 |
3 |
7.96 |
7.95 |
7.88 |
8.13 |
8.11 |
8.08 |
8.14 |
7.94 |
7.93 |
7.73 |
8.02 |
8.08 |
8.00 |
0.13 |
1.51 |
4 |
9.39 |
9.21 |
9.16 |
9.26 |
9.52 |
9.66 |
9.72 |
9.69 |
9.54 |
9.64 |
9.39 |
9.45 |
9.48 |
0.21 |
2.01 |
5 |
10.51 |
10.40 |
10.47 |
10.46 |
10.42 |
10.28 |
10.26 |
10.39 |
10.55 |
10.73 |
10.83 |
10.86 |
10.51 |
0.20 |
1.88 |
6 |
11.77 |
11.86 |
11.80 |
11.78 |
11.67 |
11.59 |
11.51 |
11.65 |
11.68 |
11.63 |
11.57 |
11.72 |
11.68 |
0.23 |
0.86 |
Mean |
0.20 |
2.22 |
* Measurement of MR of the experimental platform under different densities, and ρ1=149 kg·m-3; ρ2=155 kg·m-3; ρ3=162 kg·m-3; ρ4=170 kg·m-3; ρ5=178 kg·m-3; ρ6=187 kg·m-3; ρ7=196 kg·m-3; ρ8=207 kg·m-3; ρ9=219 kg·m-3; ρ10=233 kg·m-3; ρ11=249 kg·m-3.
We hope reviewer 2 will be satisfied with the revised manuscript.
We thank reviewer 2 again for his/her valuable suggestions and comments that have helped us improve the quality of this manuscript. We hope reviewer 2 will be satisfied with our clarification and modification.

Reviewer 3 Report
1 It is necessary to present the physical and mechanical properties the cotton variety being studied. 2 It is necessary to point out the shortcomings of the known software provision for determining the moisture content of cotton during harvest. 3 It is necessary to present the instruments and equipment used in the study, as well as errors in their measurements. 4 It is necessary to present the architecture of the neural network used in the study. 5 It is necessary to indicate the error of the model for predicting the moisture content of cotton seeds.Author Response
Response to Reviewer 3 Comments
Title: Resistive Sensing of Seed Cotton Moisture Regain Based on Pressure Compensation
Manuscript ID: sensors-2606782
Authors: Liang Fang, Ruoyu Zhang, Hongwei Duan, Jinqiang Chang, Zhaoquan Zeng, Yifu Qian, Mianzhe Hong
This is the authors’ reply to the comments and suggestions made by the reviewer 3 of the original manuscript entitled “Resistive Sensing of Seed Cotton Moisture Regain Based on Pressure Compensation”. First of all, we would like to thank the reviewer 3 for volunteering his/her valuable time in handling the review of our submission. We highly appreciate the reviewer 3’ comments and constructive remarks. This submission has been carefully revised based on the suggestions made by reviewer 3, which major changes are highlighted in red. We hope reviewer 3 will be satisfied with the revised manuscript. In the following, please kindly find the point-to-point responses.
Reviewer 3: “(1) It is necessary to present the physical and mechanical properties the cotton variety being studied.”
Authors’ Reply: We thank reviewer 3 for his/her valuable comments and suggestions on the manuscript. According to reviewer 3’s comments, in the revised manuscript, we have added the physical and mechanical properties of the studied cotton varieties [lines 94 to 96].
The revised manuscript [lines 94 to 96] now reads as follows:
Machine-harvested cotton (cultivar Xinluzao 80) harvested in Jimusaer County, Changji, Xinjiang Uyghur Autonomous Region, China (44°30′N, 88°89′E) in November 2022 were collected. The lint weight of the cotton is 5.6 g, with a lint percentage of 43.0%. The average length of the fibers in the upper half is 30.0 mm, with a strength (tenacity) of 31.4 cN/tex and a micronaire value of 4.7. The uniformity index is 85.3%[24].
We hope reviewer 3 will be satisfied with the revised manuscript.
Reviewer 3: “(2) It is necessary to point out the shortcomings of the known software provision for determining the moisture content of cotton during harvest.”
Authors’ Reply: We thank reviewer 3 for his/her valuable comments and suggestions on the manuscript. According to reviewer 3’s comments, in the revised manuscript, we have discussed the known software provisions and highlight their shortcomings. We also emphasized the importance of considering factors such as cotton density changes in order to improve the accuracy of moisture content measurement [line 76 to 79].
The revised manuscript [line 76 to 79] now reads as follows:
In summary, there is limited research on the measurement of seed cotton MR during the harvesting operations, and most studies have not considered the impact of changes in cotton density on measurement results. This greatly affects the measurement accuracy.
We hope reviewer 3 will be satisfied with the revised manuscript.
Reviewer 3: “(3) It is necessary to present the instruments and equipment used in the study, as well as errors in their measurements.”
Authors’ Reply: We thank reviewer 3 for his/her valuable comments and suggestions on the manuscript. According to reviewer 3’s comments, in the revised manuscript, we have included a detailed description of the instruments and equipment used in the Materials and Methods section [line 101 to 114; line 143 to 145].
The revised manuscript [line 101 to 114] now reads as follows:
Secondly, the cotton samples were placed in the constant temperature and humidity test chamber (FQY/WSK-400C, Guangzhou Fengqianyuan Environmental Test Equipment Co., Ltd.), with a temperature deviation of ±2.0 ℃ and a humidity deviation of ±3.0 %RH. And balance for 24 hours at different temperatures and relative humidity levels, to prepare cotton samples with different MR levels (Table 1). After the cotton samples reached equilibrium, 150 g and 50 g of cotton samples were separately weighed by an electronic balance (XY500C,Changzhou Xingyun Electronic Equipment Co., Ltd), with a measurement accuracy is ±0.01g. The 150 g cotton sample was placed in the cotton compressing device for MR measurement. The 50 g cotton sample was used to measure the actual MR value according to the "Test Method for Moisture Regain of Raw Cotton - Oven Method" [25] using the semi-automatic constant-temperature oven (YG767, Nantong Sansi Electromechanical Science & Technology Co., Ltd.), with a temperature control accuracy is ±1 ℃. This actual value was used to calibrate the MR values obtained through the resistive sensing.
The revised manuscript [line 143 to 145] now reads as follows:
The resistance of the seed cotton was measured using the bridge tester (LCR-8110G, Good Will Instrument Co., Ltd.), with a measurement accuracy is ±0.1%, and the instrument was calibrated using a standard resistor.
We hope reviewer 3 will be satisfied with the revised manuscript.
Reviewer 3: “(4) It is necessary to present the architecture of the neural network used in the study.”
Authors’ Reply: We thank reviewer 3 for his/her valuable comments and suggestions on the manuscript. According to reviewer 3’s comments, in the revised manuscript, we have provided a detailed description of the neural network architecture, including the type and size of layers, activation functions, and any other relevant details. In addition, we will provide the basic principles for selecting this specific architecture and explain how it aligns with our research objectives [line 390 to 404].
The revised manuscript [line 390 to 404] now reads as follows:
BPNN is a type of three-layer feedforward structure. The three layers are the input layer, hidden layer, and output layer (Figure 13). The input layer receives information (resistance values and pressure values) from external sources and passes it to the network for processing. The hidden layer receives information from the input layer and processes all the information. The output layer receives the processed information from the network and sends the resulting outputs to external receptors. The input signals are modified by interconnected weights known as weight factors vmn, which represent the interconnections from the mth node of the first layer to the nth node of the second layer. These modified signals are then adjusted using the tanh transfer function to compute the total activation. Similarly, the output signals from the hidden layer are adjusted by the interconnection weights wij from the ith node of the output layer to the jth node of the hidden layer. The adjusted signals are summed using the tanh transfer function, and the resulting outputs are collected at the output layer.
Figure 13. BPNN architecture diagram of seed cotton moisture regain detection model
We hope reviewer 3 will be satisfied with the revised manuscript.
Reviewer 3: “(5) It is necessary to indicate the error of the model for predicting the moisture content of cotton seeds.”
Authors’ Reply: We thank reviewer 3 for his/her valuable comments and suggestions on the manuscript. We have already specified the Mean Absolute Error of the predictive model for cotton moisture regain rate in the original manuscript [line 410 to 411]. Furthermore, we have visualized the error in the form of images [Figure 14]. We hope reviewer 3 will be satisfied with the revised manuscript.
We thank reviewer 3 again for his/her valuable suggestions and comments that have helped us improve the quality of this manuscript. We hope reviewer 3 will be satisfied with our clarification and modification.

Round 2
Reviewer 2 Report
Suggest to be accepted. Hope to make it a reality.